# Psychometric Investigation of the Workplace Social Courage Scale (WSCS): New Evidence for Measurement Invariance and IRT Analysis

**DOI:** 10.3390/bs12050119

**Published:** 2022-04-20

**Authors:** Paola Magnano, Palmira Faraci, Giuseppe Santisi, Andrea Zammitti, Rita Zarbo, Matt C. Howard

**Affiliations:** 1Faculty of Human and Social Sciences, Kore University, 94100 Enna, Italy; palmira.faraci@unikore.it (P.F.); rita.zarbo@unikore.it (R.Z.); 2Department of Science of Education, University of Catania, 95124 Catania, Italy; gsantisi@unict.it (G.S.); andrea.zammitti@phd.unict.it (A.Z.); 3Mitchell College of Business, University of South Alabama, Mobile, AL 36688, USA; mhoward@southalabama.edu

**Keywords:** courage, social courage, measurement invariance, IRT, psychometrics

## Abstract

This study, after presenting a review of the existent literature on courage and social courage in the workplace, has the purpose of providing new evidence about the psychometric properties of an Italian-language version of the Workplace Social Courage Scale (WSCS), verifying its measurement invariance across gender and the discrimination properties of its items through IRT analysis. The aim of the research is testing the Italian version of the WSCS; for this scope, four studies have been conducted on four different samples analyzing the factorial structure, the internal consistency, the measurement invariance across gender, and the convergent and concurrent validity. The results support the psychometric properties in terms of factor structure, reliability, validity, and utility, showing positive relationships with the criterion variables: satisfaction of work-related basic needs, prosocial rule breaking and work performance. The current study extends prior findings by providing further insights about the construct of courage and social courage in the workplace, especially in the Italian context. As, to date, little is known about the impact of social courage on work and organizational outcomes, the availability of a reliable, valid, and cross-culturally supported instrument can promote the role of this construct in positive organizational behavior research.

## 1. Introduction

During the course of one’s life, anyone has had to deal with adverse and uncertain conditions, periods of transition, change and growth, painful events and problematic situations, contexts in which being able to make decisions is a very difficult task. In these circumstances, dominated by doubts and unpredictability, where certainty leaves space for unknown, security for precariousness or threat, mixed feelings such as fear and courage come forward. For these reasons, Rate et al. [1] consider courage to be the intentionality of action aimed at achieving a noble purpose, even in the presence of objective risk and feelings of fear. Challenging difficulties with courage do not mean acting unconsciously but thinking about the consequences of one’s actions, pondering the different possible alternatives and taking the risk related to them.

Courage can be acted out in different contexts, such as everyday life [2], academic [3], military [4], family, social and workplace contexts [5]. The focus of this study is a particular configuration of courageous actions in the workplace that Howard et al. [6] called workplace social courage, defining it as “an (a) intentional, (b) deliberate, and (c) altruistic behavior that (d) may damage the actor’s esteem in the eyes of others” (p. 1). These authors demonstrated that workplace social courage is significantly related to personal and work outcomes, emphasizing its importance for both research and practice. Confirming his previous study, Howard [7] further underlines that social courage has positive relationships with organizational citizenship behaviors (OCBs), voice and well-being outcomes as well as negative relationships with counterproductive work behaviors (CWBs), stress, anxiety and depression.

While research on social courage is growing, certain limitations prevent its further growth. Howard et al. [6] supported the psychometric properties and validity of their measure, but these attributes have not been replicated. Researchers may be interested in studying social courage, but they may also be uncertain regarding the validity of the sole scale to measure the construct. Furthermore, only the English-language version of the measure has been investigated. Researchers in countries without English as a prevalent language are largely unable to study social courage, preventing the study of the construct for much of the modern world.

With the purpose of overcoming this gap—the lack of a validated instrument to assess workplace social courage in the Italian language—the current article creates an Italian-language version of the WSCS. The studies presented aim to investigate the psychometric properties and validity of this translated scale. Specifically, we assess the Italian WSCS’s internal consistency, factor structure, measurement invariance across gender, as well as convergent and concurrent validity.

Despite the fact that a previous study [8] established that the WSCS does not have a significant relationship with gender or femininity-masculinity, this result does not guarantee, however, that the WSCS functions similarly across genders; to address this possible concern, we verify the measurement invariance of the WSCS across gender to ensure that it is applicable in detailing the relations of workplace social courage for a sufficient range of participants.

In assessing the scale’s validity, we analyze the relation of social courage with prosocial rule breaking behaviors (PSRB), satisfaction of basic needs, and performance—each of which provide substantive insights into the nature of social courage itself.

The relationship between social courage and well-being outcomes has been explored through the framework of Self-Determination Theory (SDT; [9,10,11,12]), which is one of the most popular frameworks in explicating the antecedents and the mechanisms of subjective well-being, but it has yet to be applied in empirical research to understand courage—whether within or outside the workplace. Therefore, we propose the first attempt to understand the relation between social courage and well-being outcomes, verifying whether SDT is an effective framework for detailing the effects of social courage which can encourage future authors to utilize the framework for subsequent investigations into courage.

In sum, after a literature review on the construct of social courage in the workplace, and its relationship with positive working outcomes, we propose four studies designed to assess the psychometric properties (factor structure, internal consistency, reliability, measurement invariance across gender and validity) of the Italian version of the WSCS.

## 2. Literature Review

### 2.1. Review of the Existing Measures of Courage and Workplace Social Courage

Various types of courage have been proposed in the literature, and many attempts have been made to develop scales that can measure courage [13,14,15,16,17]. For example, Konter and Ng [18] developed the Sport Courage Scale (SCS), which evaluates courage in the sports field. Woodard and Pury [5] modified a scale originally created by Woodard [17] to develop the Woodard-Pury Courage Scale 23 (WP-23). Norton and Weiss [19] developed the Courage Measure (CM), which is composed of 12 items that evaluate courage as persistence despite fear. There is a shorter version of this scale [20] and an Italian validation [21]. Many important insights into courage were derived from these scale development efforts, but notable concerns were often observed in the measures (e.g., poor psychometric properties, questionable validity evidence) [5,20]. Subsequent authors suggested that a possible cause of these concerns was the focus on global or comprehensive measures of courage, and instead authors may benefit from developing scales for specific courage dimensions (e.g., social courage) [6,22]. One such courage dimension is moral courage, but disagreements still exist regarding its definition and measurement. Sekerka, Bagozzi, and Charnigo [23] believe that moral courage is a managerial competence; while other authors [24] point out that it is a skill that can be observed when individuals face an ethical challenge. The Professional Moral Courage scale (PMC scale; [23]) is a five-dimensional scale which includes the dimensions of moral agency, multiple values, endurance of threats, going beyond compliance, and moral goals. The PMC scale includes a single second-order factor, labelled professional moral courage, but it is still unclear whether the PMC scale is truly representative of moral courage given present disagreements regarding its construct definition.

More recently, Howard and Reiley [22] developed the Physical Courage at Work Scale (PCWS). Physical courage is defined as “a courageous behavior in which the risks involved are to the actor’s physical well-being” [22] (p. 81). The authors suggest that courage is not a unidimensional construct [6,24] and the PCWS was shown to positively relate to organizational citizenship behaviors and social courage [22]. Despite these advancements, Howard and Reiley [25] also recognized that physical courage may not be relevant to all workplace environments, and instead noted that other courage dimensions may be beneficial more broadly.

One such courage dimension relevant to most workplaces is social courage. Howard and colleagues [6] created a scale for assessing social courage. The Workplace Social Courage Scale (WSCS) consists of 11 items on a 7-point Likert-type scale (1 strongly disagree to 7 strongly agree) and showed satisfactory psychometric properties in the validation study [6]. The scale has been used in several following studies to further investigate the nature of social courage. For example, Howard & Cogswell [26] used the WSCS to investigate the antecedents of social courage. The WSCS was also used to understand the relationships between social courage, workplace outcomes, and well-being outcomes [7] and to test the role of gender in social courage [12]. In all these studies the WSCS has demonstrated satisfactory psychometric properties and relations with associated outcomes. Among its most important relations was with prosocial rule breaking (PSRB), need satisfaction, and performance—which we further review below.

### 2.2. Workplace Social Courage and Prosocial Rule Breaking Behaviors

Conceptualizations of prosocial behavior typically refer to individuals going above and beyond the specific responsibilities that are assigned to them in an effort to aid others [27,28]. While prosocial behaviors are typically normative, Morrison [29] introduced the construct of PSRB to refer to rule breaking not motivated by negative intentions toward the organization. As highlighted by Bryant et al. [30], Morrison started from the concept of rule-breaking—conceptualized as a deviant or counterproductive workplace behavior deriving from employee hostility [31], social exclusion [32] or job dissatisfaction [33]—and defined PSRB as “any instance where an employee intentionally violates a formal organizational policy, regulation, or prohibition with the primary intention of promoting the welfare of the organization or one of its stakeholders” [29] (p. 6). This conceptualization legitimates the violations of organizational rules and policies if they are functional to the achievement of the scopes of the organization. In fact, PSRB is related to positive organizational outcomes [28,29,34] but, as Howard and colleagues [6] argued, employees may be reluctant to perform these behaviors because going against organizational policies implies the possibility of conflict with supervisors or co-workers. In their study, Howard and colleagues found that social courage, which relates to positive behaviors despite social risk, is positively related to PSRB behaviors.

### 2.3. Workplace Social Courage and Satisfaction of Work-Related Basic Needs

According to Self-Determination Theory (SDT; [8,9,10,11] well-being and psychological health are maintained by the satisfaction of three basic psychological needs: (1) the need for autonomy, (2) the need for competence, and (3) the need for relatedness. The satisfaction of these needs is positively related to job resources [35,36,37,38] performance [39,40], job crafting [41,42,43] and organizational citizen behaviors [43,44]; moreover, need satisfaction is negatively related with negative affect, strain, and burnout [37,43,45]. However, some research has obtained conflicting results; for example, the need for competence was positively related to absenteeism and turnover intentions [43]. For this reason, Colledani et al. [46] highlight the need to propose new hypotheses and deepen research surrounding SDT.

The need for autonomy is defined as the desire to make one’s own choices following one’s freedom and will [8]. Some research has shown that work autonomy has an important role on in experience of stress, especially role stress, which refers to the inconsistencies of expectations associated with a role [47,48,49]. Specifically, autonomy can reduce stress [50]. Schilpzand, Hekman and Mitchell [51] developed a model of courage in the workplace based on qualitative research. Participants reported that some characteristics suggested that courageous action was appropriate, and among them was perceived autonomy. On the contrary, those who experienced low levels of perceived autonomy felt inhibited or worried about taking courageous action in the workplace.

The need for competence refers to the desire to develop new skills [8]. The need for competence also indicates the propensity to explore one’s environment and engage in behaviors to extend one’s skills. Courage allows for the determination to achieve one’s goals [52], which can produce feelings of competence. The need for relatedness represents the desire to experience closeness and connection with others [8,53]. Workplace relationships can have strong effects on individuals [54,55]: for example, good workplace relationships are connected with achieving goals [56], they favor learning and, consequently, performances [57]. In a recent study, Howard and Cogswell [26]) showed that social support in the workplace is positively related to behavioral social courage, indicating that courage is indeed associated with satisfying the need for relatedness.

### 2.4. Workplace Social Courage and Performance

Some empirical research supports the connection between social courage and various work outcomes including performance. Individuals faced with ethical challenges in the workplace can consciously and deliberately decide to act courageously [58]. When this occurs, according to Howard [7], socially courageous individuals persist in risky social situations to receive positive personal and organizational outcomes, such as performance. For the author, those who are endowed with courage in the workplace can, through a motivation approach, act and consider the positive aspects of the action rather than remain helpless. From this perspective, therefore, it would seem that courage in the workplace is a stimulus to action.

Regarding academic contexts, it has been shown that, although a confidence orientation was more adaptive than courage, it can still be considered an effective response against fear and to predict performance [3]. Courageous individuals struggle with more determination to achieve their goals [6,59], and research of executive character strengths has shown that those who act with executive integrity, bravery, and social intelligence have long-term success, both individually and in terms of the organization [60]. Furthermore, Palanski et al. [61] demonstrated the important role of courageous behavior in mediating the effects of integrity on both executive performance and image. Finally, comparing employees at different organizational levels, Tkachenko et al. [62] highlighted that the effects of behavioral courage on job performance did not vary by organizational level.

## 3. Aims of Research

Based on the review of the existing literature, workplace social courage has been found to be significantly related to personal and working outcomes such as prosocial rule breaking, satisfaction of basic needs and performance. Moreover, lacking an Italian-language measure of workplace social courage, the study aims to test the psychometric features of the WSCS when translated to Italian, verifying its measurement invariance across gender and the discrimination’s properties of its items through the IRT analysis. As four studies have been conducted to evaluate the psychometric properties of the WSCS, we have first presented each study separately, discussing the overall results at the end of the four studies. Study 1 examines the WSCS dimensionality and its factorial structure. Study 2 tests the WSCS measurement invariance by gender. Study 3 investigates the IRT parameterization of the WSCS. Study 4 examines the WSCS’s convergent and concurrent validity.

## 4. Procedure

All studies utilized the same methodological approach, and therefore it is only described here. The recruitment of the participants used convenience sampling; the participants were recruited on a voluntary basis, and they were free to abandon their participation at any moment. The data have been collected through an online survey which was approved by the ethical commission of the universities involved, and the research followed the ethical rules of the Italian Psychological Association.

### 4.1. Study 1: Scale Dimensionality and Factorial Structure

The goal of Study 1 is to examine both the factor structure and internal consistency of the Italian WSCS.

#### 4.1.1. Participants

##### Sample 1

This study consisted of 130 participants (52.3% female; M_age_ = 40.62, SD_age_ = 13.28). They were married (59.7%), unmarried (35.7%), and separated (4.7%). Most had a high school degree (52.3%) or university degree (32.8%). The remaining had a middle school degree (14.8%). They were workers in public (51.2%), private (42.6%) and non-profit (6.2%) organizations. They were working in their present organization for less than a year (14%), one to five years (25.6%), 5–10 years (5.4%), and for more than 10 years (55%).

##### Sample 2

This sample was comprised of 380 participants (57.7% female; M_age_ = 41.55, SD_age_ = 13.38). They were married (48.9%), unmarried (41.4%), and separated (9.6%). Most had a high school degree (44.1%) or university degree (48.4%). The remaining had a middle school degree (7.4%). They were workers in public (35%), private (59.9%) and non-profit (5.1%) organizations. They were working in their present organization for less than a year (9%), one to five years (29.5%), 5–10 years (12%), and for more than 10 years (49.5%). This sample was also used in Study 4 to examine the WSCS’s validity and utility.

#### 4.1.2. Measures

##### Workplace Social Courage Scale (WSCS)

This was administered in all the samples of the four studies, and therefore it is only described here. The WSCS [6] is composed of eleven items with a 7-point Likert scale, from 1 (totally disagree) to 7 (totally agree). A sample item is: “If I thought a question was dumb, I would still ask it if I didn’t understand something at work”. The scale was translated to Italian trough a back translation process, wherein the researchers agreed on the final version of the translated scale.

#### 4.1.3. Data Analysis and Results

We performed an exploratory factor analysis (EFA) using a principal axis factoring method with direct oblimin rotation on Sample 1. Based on both the resulting Scree plot and Kaiser criterion, the EFA suggested one underlying factor. The conducted parallel analysis also supported a one-dimensional solution, explaining 32.06% of the total variance. All but one item factor loading exceeded the required cut-off of 30. As a result, item 1 “Although it may damage our friendship, I would tell my superior when a coworker is doing something incorrectly” (factor loading = 0.25) was removed. Therefore, in line with original study findings, the results indicated the one-dimensionality of the scale. However, the unique emerged factor is composed of 10 items with a factor loading range between 0.680 and 0.312. The 10-item scale showed a good internal consistency. Cronbach’s alpha coefficient was 0.75, with corrected item-total correlations ranging between 0.281 and 0.566. Sample 2 was used to verify these results through a confirmatory factor analysis (CFA) with ML Robust estimation. Most items presented item loadings above 0.35, but two items had standardized item loadings below 0.30. Hence, item 4 and item 10 were eliminated. Next, based on modification indices analysis, a correlation between item 6 and item 8 residuals was added. Lastly, the following model fit results were obtained: CFI = 0.92, GFI = 0.96, SRMR = 0.05, RMSEA = 0.07, and df/X^2^ = 2.75, and NNFI = 0.91. Overall, these findings are in line with the original study. Also, a closeness to one-dimensionality assessment [63] suggested that data can be treated as essentially unidimensional, as supported by the following values: UniCo (Unidimensional Congruence) = 0.93 [BC Bootstrap 95% Confidence Interval = (0.892 − 0.961)], I-ECV (Item Explained Common Variance) = 0.84 [BC Bootstrap 95% Confidence Interval = (0.801 − 0.878)], MIREAL (Mean of Item REsidual Absolute Loadings) = 0.27 [BC Bootstrap 95% Confidence Interval = (0.236 − 0.298)]. From both EFA and CFA one single latent factor emerged, supporting the one-dimensionality of the final eight-item scale. Means, standard deviations, standardized factor loadings, standardized residuals, and corrected item-total correlations are included in Table 1.

Furthermore, factor scores for the graded data were computed based on Bayes expected a-posteriori (EAP) estimation of latent trait scores [63] with satisfying results: EAP reliability = 0.95, Factor Determinacy Index = 0.98. The strength and quality of the solution beyond pure model-data fit was also assessed by the Generalized H (G-H) index of construct replicability [64]. The H index evaluates how well a set of items represents a common factor. High H values (>0.80) suggest a well-defined latent variable, which is more likely to be stable across studies. Our findings produced satisfactory results: H-Latent = 0.86, [BC Bootstrap 95% Confidence intervals (0.815 − 0.882)], showing how well the factor can be identified by the latent response variables that underlie the observed item scores. Finally, the scale showed a satisfactory internal consistency reliability also using Sample 2 (Standardized Cronbach’s alpha = 0.84; McDonald’s omega = 0.84; greatest lower bound to reliability = 0.89). Quality and effectiveness of factor score estimates were evaluated by calculating the Factor Determinacy Index (FDI), EAP marginal reliability, and Sensitivity ratio (SR), as the number of different factor levels than can be differentiated on the basis of the factor score estimates, and expected percentage of true differences (EPTD), as the estimated percentage of differences between the observed factor score estimates that are in the same direction as the corresponding true differences. If factor scores are to be used for individual assessment, FDI values above 0.90, marginal reliabilities above 0.80, SR above 2, and EPTDs above 90% are recommended [63]. Outcomes are reported in Table 2. Conditional reliability’s function reports the statistical information corresponding to each score level of the latent trait. It is interpreted as the test information function in the IRT context. The graphic shows the conditional EAP/ORION reliability distribution, with the cut-off value of 0.80 as a vertical dotted line, against the factor standardized score estimates (M = 0.00; SD = 1.00) as “*” marks (Figure 1). The reliability of the scale is well-suited for low and central levels of social courage, which are those that can be used to detect deficiencies in the work organization. Together, these cumulative results support the psychometric properties of the Italian WSCS.

### 4.2. Study 2: Measurement Invariance across Gender

Study 2 tests the measurement invariance of the scale across gender.

#### 4.2.1. Participants

Male and female subsamples derived from sample 2 were used for this analysis. The male subsample consisted of 162 workers (M_age_ = 42.58, SD_age_ = 13.61). They were married (53.8%), unmarried (39.7%), and separated (6.4%). Most had a high school degree (47.5%) or university degree (43.1%). The remaining had a middle school degree (9.4%). They were workers in public (31.6%), private (61.4%) and non-profit (7%) organizations. They were working in the present organization for less than a year (3.8%), one to five years (32.7%), 5–10 years (8.8%), and more than 10 years (54.7%). The female subsample consisted of 218 workers (M_age_ = 40.78, SD_age_ = 13.15). They were married (45.4%), unmarried (42.6%), and separated (12%). Most had a high school degree (42.1%) or university degree (51.9%). The remaining had a middle school degree (6.1%). They were workers in public (37.4%), private (58.9%) and non-profit (3.7%) organizations. They were working in the present organization for less than a year (13%), 1–5 years (27%), 5–10 years (14.4%), and more than 10 years (45.6%).

#### 4.2.2. Measure

The WSCS, as described in Section Workplace Social Courage Scale (WSCS) was administered.

#### 4.2.3. Data Analysis and Results

A multigroup confirmatory factor analysis (MGCFA; robust maximum likelihood estimation) was performed to assess the factor invariance of the Italian WSCS in the two male-female subsamples. We only included the items retained from Study 1 in our assessment of the Italian WSCS in Studies 2, 3, and 4. In the hierarchical ordering of nested models, constraints were added to each model. The fit of four nested models was analyzed. In addition, the comparative fit of two nested models was assessed. Statistical differences in fit indexes were measured to establish the highest level of invariance. First, the empirical model was analyzed to verify that all items presented item loadings above 0.30. Then, based on a modification indices analysis, correlations between items with the higher standardized residuals were added in a stepwise manner until no modification indices above 10 remained. This resulted in the addition of four residual correlations (between item 6–8, item 3–11, item 2–8, item 7–9). All these item pairs seemed conceptually related. The goodness of fit indexes for the configural invariance highlighted the equivalence of the basic measurement models: CFI = 0.97, GFI = 0.97, and RMSEA = 0.05, meaning that males and females conceptualize workplace social courage in the same way. The metric invariance model was configured by adding constraints to the factor loadings to the basic model in order to make the factor loadings (Table 3) equal across groups. Although the SB χ^2^ value was statistically significant (*p* < 0.05), the model’s fit seemed acceptable due to the results referring the alternative indexes, indicating equality in the WSCS’s measurement intervals across groups. Also, the scalar invariance model showed a good fit; no significant discrepancy was observed between scalar and metric invariance models, ΔSB χ^2^ (7) = 12.31, *ns*, and their respective CFIs did not exceed 0.01. Thus, these outcomes imply that both groups share origins. The residual invariance model also showed an acceptable fit (no statistically significant differences were found in the SB χ^2^ test, ΔSB χ^2^ (10) = 18.52, *ns*), meaning that the workplace social courage measure is similar in both subsamples. The fit indexes for the four invariance factor models are reported in Table 4. Table 3 shows the standardized factor loadings (λ) for the four invariance models.

### 4.3. Study 3: IRT Discrimination

Study 3 is aimed at investigating the item response theory parameterization to highlight the pattern of item discriminations through the discrimination parameter (a_j_), indicating the quality of the item as a measure of the construct, and the difficulty parameter (b_j_), indicating the item location.

#### 4.3.1. Participants

Sample 3 consisted of 450 participants (48% female; M_age_ = 40.05, SD_age_ = 12.33). They were married (55.8%), unmarried (37.9%), and separated (6.3%). Most had a high school degree (55.8%) or university degree (6.3%). The remaining had a middle school degree (37.9%). They were workers in public (53.4%), private (11.4%) and nonprofit (35.2%) organizations. They were working in the present organization for less than a year (12.1%), one to five years (29.1%), 5–10 years (15.5%), and more than 10 years (43.3%).

#### 4.3.2. Measure

The WSCS as described in Section Workplace Social Courage Scale (WSCS) was administered.

#### 4.3.3. Data Analysis and Results

Given that the unidimensional assumption was met in the prior analyses, the WSCS was further analyzed using Rasch models that are suitable when the scale is unidimensional. Item 5 showed infit greater than 1.5 (1.74), as did item 8 (1.82). Although these items were beyond the cutoff of 1.50, according to several studies, MNSQ values smaller than 2 are not significantly detrimental to measurement [65] Further items reported fit indexes between 0.69 and 1.50, verifying the model requirements (see Table 5). Item response theory parameterization by the normal-ogive graded response model (GRM) was estimated to highlight the pattern of item discriminations in the overall scale. Two types of parameters were analyzed: (1) the discrimination parameter aj, indicating the quality of the item as a measure of the trait; the higher a_j_ is the more precise and informative the item is about the trait that is measured, and (b) the difficulty b_j_ parameter, indicating the item location, which is the point on the ability scale at which the probability of endorsing the item is 0.50. Table 6 shows findings from the item response theory parameterization by normal-ogive graded response model (GRM). Specifically, the pattern of item discriminations was highlighted reporting estimates for parameters a_j_ (item discriminability), and parameters b_j_ (item endorsement). No item was observed with a low or very low level (0.21 < a_j_ < 0.40 or a_j_ < 0.20), either with moderately discriminating 0.41 < a_j_ < 0.80 or high discrimination 0.81 < a_j_ < 0.1. Thus, item discrimination can be considered as heavily adequate since all items revealed very high discrimination (a_j_ ˃ 1). As regards the item location, the trends of the mean location scores should increase as the categories scores increase. From this analysis it can be observed that all items have an increasing mean ability score as the categories scores increased. At that point the theoretical expectation has been fulfilled. Overall, WSCS items showed low and medium b estimated parameters, displaying relatively low difficulty (−3.44 < b_2_ < 0.66, −3.03 < b_3_ < 0.49, −3.03 < b_5_ < 0.73, −2.87 < b_6_ < 0.20, −3.23 < b_7_ < 1.81, −2.69 < b_8_ < 0.42, −3.51 < b_9_ < 1.96, −4.43 < b_11_ < 0.65), thus indicating a tendency of the items to require a small amount of workplace social courage to be endorsed and less ability to discriminate among high levels of the latent variable. According to our results, the item discriminability parameters have shown a good discrimination power, and the item endorsement parameters have shown relatively low difficulty. As a consequence, it seems that the WSCS items discriminate well only between people that score low and medium in social courage, whereas it may not discriminate as well between people high in the construct. Future research might consider the possibility of introducing additional items to improve their distribution over the construct, providing a measure able to cover also the upper end of the spectrum.

### 4.4. Study 4: Validity and Utility

The aim of Study 4 is to examine the convergent validity of the Italian WSCS. To gather convergent validity evidence, the scale should be positively related to existing courage measures [6]; social courage should also be related to relatedness as a specific aspect of work-related basic needs satisfaction. Lastly, study 4 investigates the scale’s utility in terms of significant relationships between social courage and important workplace outcomes (i.e., prosocial rule breaking and performance). To this aim we formulated the following hypotheses:

**Hypothesis** **1** **(H1).**
*Social courage is positively related to prosocial rule breaking.*


**Hypothesis** **2** **(H2).**
*Social courage is positively related to performance.*


**Hypothesis** **3** **(H3).**
*Social courage is positively related to autonomy need satisfaction.*


**Hypothesis** **4** **(H4).**
*Social courage is positively related to competence need satisfaction.*


#### 4.4.1. Participants

We used the participants from sample 2: 380 workers (57.7% female; M_age_ = 41.55, SD_age_ = 13.38). They were married (48.9%), unmarried (41.4%), and separated (9.6%). Most had a high school degree (44.1%) or university degree (48.4%). The remaining had a middle school degree (7.4%). They were workers in public (35%), private (59.9%) and non-profit (5.1%) organizations. They were working in their present organization for less than a year (9%), one to five years (29.5%), 5–10 years (12%), and more than 10 years (49.5%).

#### 4.4.2. Measures

##### Workplace Social Courage Scale

The WSCS is described in Section Workplace Social Courage Scale (WSCS).

##### Courage Measure

The Italian adaptation of the Courage Measure (CM [19,21]) derives from the short version of CM proposed by Howard and Alipour [20]; it is composed of six items with a 7-point Likert scale from 1 (never) to 7 (always). A sample item is: “I tend to face my fears.” The Cronbach’s alpha of the scale was 0.85.

##### Prosocial Rule Breaking Scale

The Prosocial Rule Breaking (PRSB [34]) is a 13-item scale answered on a 5-point Likert scale, from 1 (completely disagree) to 5 (completely agree). A sample item is: “I break organizational rules or policies to do my job more efficiently”. The scale was translated in Italian through a backtranslation process conducted by two bilingual researchers. The Cronbach’s alpha calculated on the whole scale for the study sample was 0.91.

##### Performance

Work performance was measured with a four-item scale [66] answered on a 5-point Likert scale, from 1 (very bad) to 5 (very good). The Cronbach’s alpha calculated on the sample of the study is 0.76.

##### Work-Related Basic Need Satisfaction Scale

The Work-related Basic Need Satisfaction Scale (WBNS [37]; Italian adaptation by Colledani et al. [46]) is composed of eighteen items (six for each basic need), answered on a 5-point Likert scale (from 1 = completely disagree to 5 = completely agree). The three dimensions analyzed are: Competence, which describes a feeling of effectiveness (a sample item is: “I have the feeling that I can even accomplish the most difficult tasks at work”); Autonomy, which refers to a sense of volition and psychological freedom (a sample item is: “The tasks I have to do at work are in line with what I really want to do”); Relatedness, which conveys the feeling of being loved and supported (a sample item is: “Some people I work with are close friends of mine”). The Cronbach’s alphas calculated on the study sample were: 0.80, 0.77, and 0.69, respectively.

#### 4.4.3. Data Analysis and Results

The correlations of variables administered in Study 4 are included within Table 7. Convergent validity was supported with a measure of courage. The WSCS robustly correlated with CM (r = 0.46, *p* < 0.01), showing adequate convergent validity. Next, the WSCS was expected to be related to WBNS relatedness. The WSCS had significant moderate correlations with relatedness (r = 0.25, *p* < 0.01), thus supporting the scale’s concurrent validity. Four separate hierarchical regression analyses for each dependent variable (prosocial rule breaking, performance, satisfaction’s needs for autonomy, and satisfaction’s needs for competence) were conducted to evaluate the contribution of social courage on workplace outcomes. The variables were entered into the regression analysis in two hierarchical steps. In the first step, social courage was inserted to examine its association with prosocial rule breaking, performance, autonomy, and competence in the first, second, third, and fourth regression analyses, respectively. In the second step, courage was introduced to control for possible confounding effects. The first regression analysis showed that workplace social courage predicted the prosocial rule breaking (β = 0.13, *p* = 0.01). As shown in Table 8, courage introduced in the second step accounted for a significant proportion of the explained variance (ΔR^2^ = 0.02, ΔF = 7.12, *p* = 0.008). This result shows that controlling for courage changed the observed relations between the measured variables. Indeed, courage (β = 0.15, *p* = 0.008) contributed significantly to the prediction of prosocial rule breaking, whereas social courage did not contribute significantly to the prediction of prosocial rule breaking (β = 0.06, ns). According to the second regression analysis results, workplace social courage predicted performance (β = 0.32, *p* = 0.000). As shown in Table 8, courage introduced as a control variable in the second step of the hierarchical analysis accounted for a significant proportion of the explained variance (ΔR^2^ = 0.06, ΔF = 26.08, *p* = 0.000). This result shows that controlling for courage did not change the observed relations between the variables of interest. Furthermore, courage (β = 0.27, *p* = 0.000) contributed significantly to the prediction of performance. According to the third regression analysis results, workplace social courage predicted autonomy (β = 0.16, *p* = 0.002). As shown in Table 8, courage introduced as the control variable in the second step of the hierarchical analysis accounted for a significant proportion of the explained variance (ΔR^2^ = 0.03, ΔF = 11.32, *p* = 0.001). This result shows that controlling for courage changed the observed relations between the variables of interest. Indeed, courage (β = 0.19, *p* = 0.001) contributed significantly to the prediction of autonomy, whereas social courage did not contribute significantly to the prediction of autonomy (β = 0.08, ns). The fourth regression analysis showed that workplace social courage predicted the competence levels (β = 0.35, *p* = 0.000). As shown in Table 8, courage introduced in the second step accounted for a significant proportion of the explained variance (ΔR^2^ = 0.07, ΔF = 33.01, *p* = 0.000). This result shows that controlling for courage did not change the observed relations between the variables of interest. Furthermore, courage (β = 0.30, *p* = 0.000) contributed significantly to the prediction of performance. Table 8 shows the results of the regression models.

## 5. Overall Discussion

The aim of our four studies was to test the psychometric properties of the WSCS using an Italian sample. Specifically, we sought to verify its factor structure and reliability, measurement invariance across gender, IRT parametrization, and convergent, concurrent, and predictive validity. The findings confirm the one-dimensionality of the scale after deleting three items (item 1, item 4, and item 10) that did not show the acceptable item loading thresholds. The failure of these items in the group of Italian workers can be explained by cultural reasons related to the way of thinking and acting in a working environment. Referring to item 1 “Although it may damage our friendship, I would tell my superior when a coworker is doing something incorrectly”, this behavior is not labelled as courage, but describes something as being a spy; generally, it is acted for personal interests related to competition and would never be declared. In fact, such type of behaviors in the Italian working context can be read as a defence of individual interests, rather than acting for a noble purpose in favour of the organization; moreover, denouncing the colleagues’ negligence could be an indicator of conformism or obedience or a wish to indulge the management rather than courage.

Furthermore, with regard to items 4 and 10, both referred to being available to coordinate a project that may fail or to submit a project with the risk of appearing inadequate. Generally, in Italian workplaces, we cannot rely on such sensitivity driven by a strong affiliation with the company that supports the willingness to sacrifice oneself to coordinate or present projects at risk in the name of the company.

The internal consistency reliability of the scale was satisfactory and the measurement invariance across gender was confirmed. Regarding the IRT discrimination, we have some concerns regarding the WSCS items’ discrimination capacity at all levels of the underlying construct. The WSCS items were supported to discriminate well only between people that scored low and medium in social courage, whereas it may not discriminate as well between people high in the construct. Furthermore, the convergent validity was confirmed through the positive and strong correlation with another measure of courage; the concurrent validity was demonstrated through the significant relationships with relatedness satisfaction as a component of work-related basic needs. Finally, the utility of the WSCS was investigated and confirmed in terms of relationships with workplace outcomes (prosocial rule breaking, performance, autonomy, and competence), using courage as a control variable. The observed relationship between courage and social courage was expected, as the two constructs partially overlap; more specifically, social courage is a dimension from the broader construct of courage. Previous studies have also shown similar results using the same scales [6,12,67]. Concerning the relationship between workplace social courage and relatedness need satisfaction, we have not found studies that directly explored the same constructs; however, in a recent study, Howard and Cogswell [26] supported a link between social courage and social support in workplaces. The need for relationships represents the desire to feel close connections to others, such as colleagues and co-workers in the workplace [8,53]. Social courage in the workplace is essentially an altruistic behavior that, in the long term, will have positive consequences on the quality of relationships in the workplace; this aspect could explain the link between these constructs, as courage could be identified as a source of relatedness needs satisfaction in the workplace. These relations of social courage with relatedness and autonomy also provide notably theoretical implications by linking the construct with SDT. Most research on social courage has linked the construct with the approach/avoidance framework [7,67]. While these studies have provided important insights, it is necessary to understand social courage via the lens of multiple theories to understand its nuanced relationships. By associating social courage with SDT, we highlight novel avenues that the construct may relate to well-being. Specifically, those with greater social courage may be more likely to satisfy their relatedness needs, as argued above, but they may also put themselves in positions with greater control over their surroundings (i.e., autonomy). By doing so, they can more fully satisfy their needs and experience greater psychological well-being. Now that this initial link has been observed, we call on future research to provide more in-depth studies regarding social courage and SDT. The relationship between workplace social courage and prosocial rule breaking, as underlined in the theoretical section, has been directly explored in a few recent studies [6,12], but the two constructs are linked by the underlying concept of risk and prosocial motivation. Workplace social courage implies a risk-taking propensity [29] and the violation of the organizational policies—although aimed to a functional purpose—creates risky circumstances for the workers in terms of the relationships with superiors or co-workers. Similarly, although PSRB benefits the organization, it does violate norms. Employees performing PSRB behaviors risk reprimand from their supervisors. Therefore, social courage may relate to PSRB due to two key characteristics of both concepts—risk and prosocial motivation. Social courage in the workplace has been previously related to working performance, as shown by Tkachenko and colleagues [62], who underlined the relevant role of behavioral courage in predicting job performance. Moreover, similar results were found by Palanski and colleagues [61].

## 6. Conclusions and Limitations

The current article provides an advancement in research on courage and some practical benefits. Specifically, prior research on social courage was limited to English-speaking contexts, whereas the construct can now be studied in populations that speak Italian. Given the importance of courage to the modern world, we also call on future researchers to continue translating the WSCS and other courage measures to other languages. By doing so, researchers could also observe cross-cultural differences in courage. For instance, cultures differ based on power distance norms, and it may be more daunting to perform a courageous behavior with greater power distances. A future study could assess whether courage is indeed less common in cultures with greater power distances, which could support—or not support—whether current research on courage (primarily performed in Western, English-speaking populations) can be generalized to broader populations. Further research on workplace social courage could involve specific targets, such as, for example, whistle-blowers who can be considered as an example of courageous people in organizations. In addition, further research is required to examine the role of gender in acting courageously in organizations, exploring whether the status in the organization can play a mediational or moderational effect.

The limitations of the study are mainly with regard to its cross-sectional nature and the lack of an objective outcome, which limits its predictive validity. Longitudinal research designs using at least an objective outcome—as, for example, the working performance—could strengthen the predictive validity of the scale; other studies could explore the role of workplace social courage on other organizational dimensions that require the workers to act courageously, putting at risk their image, role, or their reputation in the workplace [68].

## Figures and Tables

**Figure 1 behavsci-12-00119-f001:**
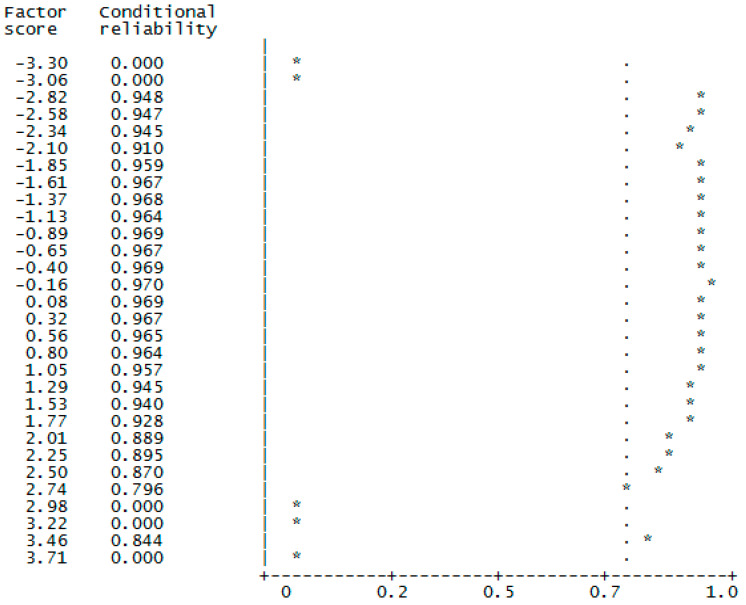
Distribution of conditional Bayes expected a posteriori estimation (EAP)/Orion reliability for the factor scores of the unidimensional WSCS.

**Table 1 behavsci-12-00119-t001:** WSCS item means, SD, standardized factor loadings, and corrected item-total correlations (Sample 2).

Item Number	Mean	SD	Standardized Factor Loadings	Standardized Residuals	Corrected Item-Total Correlation
WSCS 2	5.38	1.55	0.534	0.845	0.484
WSCS 3	5.56	1.52	0.378	0.926	0.367
WSCS 5	5.58	1.44	0.613	0.790	0.492
WSCS 6	5.79	1.40	0.691	0.790	0.551
WSCS 7	5.14	1.48	0.390	0.921	0.397
WSCS 8	5.67	1.26	0.683	0.731	0.541
WSCS 9	5.02	1.56	0.385	0.923	0.405
WSCS 11	5.31	1.40	0.571	0.821	0.475

**Table 2 behavsci-12-00119-t002:** Quality and effectiveness of factor score estimates.

Factor Determinacy Index (FDI)	0.976
EAP marginal reliability	0.953
Sensitivity ratio (SR)	4.501
Expected percentage of true differences (EPTD)	96.6%

**Table 3 behavsci-12-00119-t003:** Standardized factor loadings (λ) for invariance models.

Item	CI λ (Male)	CI λ (Female)	MI λ	ScI λ	StI λ
Item 2	0.979	0.939	0.942	0.939	0.940
Item 3	0.629	0.706	0.683	0.667	0.648
Item 5	0.799	0.856	0.839	0.839	0.846
Item 6	0.737	0.896	0.841	0.844	0.844
Item 7	0.378	0.752	0.619	0.602	0.594
Item 8	0.813	0.786	0.789	0.809	0.823
Item 9	0.411	0.885	0.757	0.726	0.658
Item 11	0.740	0.899	0.840	0.852	0.843

Note. CI = configural invariance; MI = metric invariance; ScI = scalar invariance; StI = strict invariance.

**Table 4 behavsci-12-00119-t004:** Fit indexes for invariance factor models.

Nested Models	SBχ^2^ (df), *p*	CFI	GFI	NNFI	RMSEA	SRMR	Δχ^2^ (Δdf), *p*	ΔCFI
Model 1. Configural invariance	51.70 (34), *p* < 0.05	0.97	0.97	0.95	0.05	0.05	−	−
Model 2. Metric invariance	64.05 (42), *p* < 0.05	0.96	0.96	0.95	0.05	0.07	10.04 (7), *ns*	0.01
Model 3. Scalar invariance	76.42 (49), *p* < 0.001	0.95	0.99	0.95	0.05	0.07	12.31 (7), *ns*	0.01
Model 4. Strict invariance	113.94 (60), *p* < 0.001	0.91	0.99	0.92	0.07	0.08	18.52 (10), *ns*	0.04

Note. SB χ^2^ = Satorra–Bentler scaled chi-squared test; CFI = comparative fit index; GFI = goodness-of-fit index; NNFI = Non-normed fit index; RMSEA = root mean square error of approximation; SRMR = Standardized root mean square residual.

**Table 5 behavsci-12-00119-t005:** WSCS items infit statistics.

Item	Infit	Item	Infit
Item 2	1.04	Item 7	1.37
Item 3	1.28	Item 8	1.82
Item 5	1.74	Item 9	1.26
Item 6	1.50	Item 11	1.26

**Table 6 behavsci-12-00119-t006:** IRT parameterization of the WSCS items: item discrimination and item difficulty measures with related standard errors.

	a	b 1	b 2	b 3	b 4	b 5	b 6
Item 2	1.46 ± 0.15	−3.44 ± 0.36	−2.67 ± 0.25	−1.98 ± 0.18	−1.56 ± 0.15	−0.65 ± 0.09	0.66 ± 0.11
Item 3	1.34 ± 0.14	−3.03 ± 0.31	−2.51 ± 0.24	−1.98 ± 0.19	−1.55 ± 0.15)	−0.80 ± 0.11	0.49 ± 0.10
Item 5	1.53 ± 0.15	−3.03 ± 0.29	−2.45 ± 0.22	−1.94 ± 0.17	−1.35 ± 0.13	−0.54 ± 0.09	0.73 ± 0.11
Item 6	2.08 ± 0.21	−2.87 ± 0.26	−2.27 ± 0.18	−2.04 ± 0.16	−1.63 ± 0.13	−1.01 ± 0.09	0.20 ± 0.08
Item 7	1.00 ± 0.12	−3.23 ± 0.37	−2.49 ± 0.28	−1.72 ± 0.20	−0.81 ± 0.13	0.17 ± 0.11	1.81 ± 0.22
Item 8	2.46 ± 0.25	−2.69 ± 0.22	−2.38 ± 0.18	−1.82 ± 0.13	−1.41 ± 0.10	−0.65 ± 0.07	0.42 ± 0.08
Item 9	1. ± 0.12	−3.51 ± 0.41	−2.49 ± 0.28	−1.76 ± 0.21	−0.92 ± 0.14	0.21 ± 0.11	1.96 ± 0.24
Item 11	1.45 ± 0.15	−4.43 ± 0.59	−3.51 ± 0.37	−2.35 ± 0.22	−1.60 ± 0.15	−0.73 ± 0.10	0.65 ± 0.11

Note. In parentheses bias-corrected bootstrap 95% confidence intervals for IRT parameterization. Legend. a = item discrimination; b = category difficulty.

**Table 7 behavsci-12-00119-t007:** Correlations of the WSCS testing convergent and concurrent validity.

	WSCS	CM	WBNS—Relatedness
WSCS	0.763		
CM	0.457 **	0.840	
WBNS—Relatedness	0.253 **	0.182 **	0.716

Note. Reliabilities are on diagonal; WSCS = Workplace Social Courage Scale, CM = Courage Measure, WBNS = Work related Basic Need Satisfaction; ** *p* < 0.01.

**Table 8 behavsci-12-00119-t008:** Hierarchical regression analyses.

	*β*	*t*	*R* ^2^	Δ*R*^2^	Δ*F*
*Prosocial rule breaking*					
Step 1					
Social courage	0.13	2.50 **	0.02	0.02	6.23 *
Step 2					
Social courage	0.06	1.02	0.03	0.02	7.12 **
Courage	0.15	2.67 **			
*Performance*					
Step 1					
Social courage	0.32	6.46 ***	0.10	0.10	41.74 ***
Step 2					
Social courage	0.19	3.60 ***	0.16	0.06	26.08 ***
Courage	0.27	5.11 ***			
*Autonomy*					
Step 1					
Social courage	0.16	3.19 **	0.03	0.03	10.18 **
Step 2					
Social courage	0.08	1.34	0.06	0.03	11.32 **
Courage	0.19	3.36 **			
*Competence*					
Step 1					
Social courage	0.35	7.36 ***	0.13	0.13	54.16 ***
Step 2					
Social courage	0.22	4.20 ***	0.20	0.07	33.01 ***
Courage	0.30	5.75 ***			

Note. * *p* < 0.05; ** *p* < 0.01; *** *p* < 0.001.

## Data Availability

Data will be provided by the corresponding author under reasonable request.

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
