# Peer review of "Psychometric Investigation of the Workplace Social Courage Scale (WSCS): New Evidence for Measurement Invariance and IRT Analysis"

_behavsci, 2022, doi:10.3390/bs12050119_

Round 1

Reviewer 1 Report

in the previous period, I proposed rejections of almost all manuscripts I reviewed. So I was happy to receive one acceptable manuscript eventually.

I will add several comments:

1. About the notion of "courage". Namely, nothing is always clearly black or white. Courage, especially in work settings, has definitely a "noble purpose" but only partly, and partly it could be a defense of one's interests or an indicator that the person is a "spy" as it is written in line 503. Maybe, the authors should discuss the notion a little bit more. The blame of colleagues could be also an indicator of conformism or obedience or a wish to indulge the management, rather than courage.

2. I would like to see some reference to whistleblowers who are, in my opinion, real examples of courageous people in organizations. It could give more connections to real-life situations and contribute to the additional validity of the study. 

3. WSCS and gender. It seems that mentioned differences could be attributed to some other, third factor that influences both: level of courage and gender. For instance status at work. Maybe females have lower status and therefore they do not dare to express their courage (e.g. not because of their gender but because of their stats). It could be discussed further.

4. Line 82 Is it a subtitle? if yes it deserves a clearer position. Or just an unfinished sentence?

5. Regarding the content of the scale. Authors give us only one item “Although it may damage our friendship, I would tell my superior when a co-worker is doing something incorrectly” in line 234 but only 11 lines later inform us that they reject just this item.
So, I would like to see a clearer connection between the definition of courage given at the beginning e.g. "altruistic behavior" and items from the WSCS scale. I am aware that the authors just want to "provide new evidence about the psychometric properties of an Italian-language version of the" scale, but they basically evaluate quality of the scale, and its validity is definitely, among the most important characteristic of the WSCS

Author Response

In the previous period, I proposed rejections of almost all manuscripts I reviewed. So, I was happy to receive one acceptable manuscript eventually.

Thank you very much for your positive feedback.

I will add several comments:

  1. About the notion of "courage". Namely, nothing is always clearly black or white. Courage, especially in work settings, has definitely a "noble purpose" but only partly, and partly it could be a defense of one's interests or an indicator that the person is a "spy" as it is written in line 503. Maybe, the authors should discuss the notion a little bit more. The blame of colleagues could be also an indicator of conformism or obedience or a wish to indulge the management, rather than courage.

Thank you very much for this further reflection on our comments. We have added these arguments in our discussion

  1. I would like to see some reference to whistleblowers who are, in my opinion, real examples of courageous people in organizations. It could give more connections to real-life situations and contribute to the additional validity of the study. 

Thank you for this suggestion, we hadn't thought about it. We have added it in the final paragraph

  1. WSCS and gender. It seems that mentioned differences could be attributed to some other, third factor that influences both: level of courage and gender. For instance, status at work. Maybe females have lower status and therefore they do not dare to express their courage (e.g. not because of their gender but because of their stats). It could be discussed further.

As gender differences were not explicitly discussed in the manuscript, we have added this further reflection in the conclusions, as suggestions for further research

  1. Line 82 Is it a subtitle? if yes it deserves a clearer position. Or just an unfinished sentence?

Sorry, it was a typo.

  1. Regarding the content of the scale. Authors give us only one item “Although it may damage our friendship, I would tell my superior when a co-worker is doing something incorrectly” in line 234 but only 11 lines later inform us that they reject just this item. So, I would like to see a clearer connection between the definition of courage given at the beginning e.g., "altruistic behavior" and items from the WSCS scale.

Sorry for the inconvenient: we have changed the sample item of the scale, reporting one more representative

I am aware that the authors just want to "provide new evidence about the psychometric properties of an Italian-language version of the" scale, but they basically evaluate quality of the scale, and its validity is definitely, among the most important characteristic of the WSCS

We have rephrased.

Reviewer 2 Report

  • The title and the abstract coincide with the content of the paper. Keywords are well-chosen.
  • In the abstract, I propose to emphasise the aim of the work, the research question posed or the hypothesis. The abstract must focus on objectives, mention how they were achieved, and emphasize the results obtained.
  • I propose in the introduction should specify the methodology of research and research hypotheses. The diagnosis itself should indicate the novelty of the results and to publish the considerations in scientific journals. It should define the purpose of the work and its significance, including specific hypotheses being tested. The current state of the research field should be reviewed carefully and key publications cited.
  • Unacceptable division of the article, creates confusion
  • I propose a more classical structure of the article, where we have introduction, literature research, methodology, research, discussion and conclusions (optional).
  • From part of "Workplace social courage and satisfaction of work-related basic needs, Workplace social courage and performance" the article I propose to make a literature review chapter.
  • The chapter "Aims of research" should be called Methods and Methodology
  • The main problem is the epistemological structure (why the article was conceived and how the study was developed). I suggest the following structure of objectives: (i) research gap; (ii) research question; (iii) purpose of the article; (iv) intermediate objectives  (v) assumptions or hypo; and (vi) research method. This structure must appear in the introduction.
  • Chapter results and discussion are important parts of the article.
  • In the chapter: Results-show objectively - as far as possible - the key findings of the research, but without interpreting them.
  •  

Author Response

  • The title and the abstract coincide with the content of the paper. Keywords are well-chosen.

Thank you very much for your feedback.

  • In the abstract, I propose to emphasise the aim of the work, the research question posed or the hypothesis. The abstract must focus on objectives, mention how they were achieved, and emphasize the results obtained.

We have rephrased the central part of the abstract, highlighting more clearly purposes, aims and results.

  • I propose in the introduction should specify the methodology of research and research hypotheses. The diagnosis itself should indicate the novelty of the results and to publish the considerations in scientific journals. It should define the purpose of the work and its significance, including specific hypotheses being tested. The current state of the research field should be reviewed carefully, and key publications cited.

We have rephrased the final part of the introduction to present more clearly the novelty of our study, the existent gaps in the literature on workplace social courage and the purposes of the studies reported.

  • Unacceptable division of the article, creates confusion

We are sorry for this inconvenient.

  • I propose a more classical structure of the article, where we have introduction, literature research, methodology, research, discussion and conclusions (optional).

Due to the complex nature of the empirical study, we have organized the research section as follows: we have presented the four studies separately; for each study we have applied the classical structure: participant, measures, data analysis and results. At the end of the four studies, we have presented the overall discussion of the results. We expect this structure make the manuscript more understandable.

  • From part of "Workplace social courage and satisfaction of work-related basic needs, Workplace social courage and performance" the article I propose to make a literature review chapter.

Thank you for this suggestion. We have followed it. 

  • The chapter "Aims of research" should be called Methods and Methodology

According to the new structure of the manuscript, we have leaved the title of the paragraph as it was, changing the following ones.

  • The main problem is the epistemological structure (why the article was conceived and how the study was developed). I suggest the following structure of objectives: (i) research gap; (ii) research question; (iii) purpose of the article; (iv) intermediate objectives (v) assumptions or hypo; and (vi) research method. This structure must appear in the introduction.

Following this suggestion, we have concluded the introduction according to the proposed structure

  • Chapter results and discussion are important parts of the article.
  • In the chapter: Results-show objectively - as far as possible - the key findings of the research, but without interpreting them.

It was very difficult avoiding some reasonings especially regarding the deleting of the items, as a justification was required. A part from the part just mentioned (line 515-529), the paragraph seems us to discuss the results in light of the literature 

Reviewer 3 Report

 Only minor changes are needed :

  1. Row 82. Please, make sure this sentence is a subtitle and not a paragraph
  2. Row 125. Be consistent when writing the subtitle since it is written with initial capital letters, while the rest are written in lowercase.
  3. Please, extend the Conclusion and limitations section as it is too short.

Congrats again on the paper.

Author Response

Only minor changes are needed:

  1. Row 82. Please, make sure this sentence is a subtitle and not a paragraph

Sorry, it was a typo.

  1. Row 125. Be consistent when writing the subtitle since it is written with initial capital letters, while the rest are written in lowercase.

We have corrected

  1. Please, extend the Conclusion and limitations section as it is too short.

Following your suggestion, we have improved the paragraph.

Congrats again on the paper.

Thank you very much for your appreciation

Round 2

Reviewer 2 Report

comments have been taken into account,  accept in present form